# Perceived Benefit of Immunization-Trained Technicians in the Pharmacy Workflow

**DOI:** 10.3390/pharmacy8020071

**Published:** 2020-04-21

**Authors:** Taylor G. Bertsch, Kimberly C. McKeirnan

**Affiliations:** Department of Pharmacotherapy, College of Pharmacy and Pharmaceutical Sciences, Washington State University, Spokane, WA 99202, USA; kimberly.mckeirnan@wsu.edu

**Keywords:** immunizing pharmacy technicians (IPT), community-pharmacy immunizations, assessment of workflow in community pharmacy

## Abstract

Clinical community pharmacists have continually restructured their workflow to serve the community by optimizing patient care outcomes. Defining the perceived benefits of having an immunizing pharmacy technician in the workflow can help to redefine the way community pharmacists operate during patient immunization. The purpose of this study is to share the opinions of supervising pharmacists that have an immunizing technician within their workflow model and highlight their contributions. Pharmacists involved in this novel workflow model were interviewed two times, once in 2017 and then in 2020, to gauge opinions over time. Findings in the results of this study included such themes as: (1) Pharmacists’ perceived improvement in workflow flexibility; (2) The choice of the correct technician to immunize within the pharmacy; (3) Pharmacists’ perceived improved workflow time prioritization; (4) Limited available training as a barrier to implementation; and (5) The initial apprehension and later acceptance of pharmacists with respect to the innovation. As technician immunization administration spreads beyond early adopter states, further research into the impact on pharmacy workflow is needed.

## 1. Introduction

The Department of Health and Human Services supports the expansion of the role of the pharmacist, enhancing patient autonomy and providing competition within the current healthcare model [1]. Advancements in the role of the pharmacist in community-based practice to meet the needs of patients within the area they serve have been largely successful [2]. The impact of pharmacy-based immunization services has resulted in millions of additional immunizations being given annually [3,4]. Given the increased need to provide continued vaccination efforts in the United States [5,6] and promote community-pharmacist role advancement, a transition of current workflow responsibilities could be considered in order to support this change. 

When pharmacists move from traditional dispensing roles to increasingly clinical roles, the need for pharmacy technicians to take on advanced roles increases. According to Koehler and Brown, pharmacy technician roles have historically evolved when the role of the pharmacist has changed, creating gaps and a need for technicians to perform new tasks [7]. Literature supports advancing the role of pharmacy technicians to improve patient outcomes within the pharmacy, particularly when training is available and there is a clear and tangible benefit to the technician [8]. 

In recent years, pharmacy technicians have taken on several new roles, including accepting verbal prescriptions, performing prescription transfers, and checking prescriptions. Results show these new technician roles have had a positive impact on pharmacy workflow. Fleagle and colleagues piloted a tech-check-tech program in a community pharmacy setting and found technicians were at least as accurate as pharmacists in checking prescriptions, with the potential to save pharmacists approximately 23 working days per year by performing this task [9]. A qualitative study by Hohmeier and colleagues showed that high-performing pharmacy sites had pharmacy technicians engaged in both nonclinical and clinical support activities [10]. Clinical activities at high-performing pharmacies included responsibilities such as scheduling patient appointments with pharmacists, preparing patient charts for the pharmacists prior to appointments, and documenting patient communication [10].

A recent role for pharmacy technicians in the United States includes the administration of immunizations. Technician immunization advancement has gained momentum and support since it began in Idaho in 2017. Three states legally allow pharmacy technicians to administer immunizations [11,12,13]. Eid and colleagues assessed the regulatory nature of pharmacy technician vaccine administration with a nine-question survey sent to 51 state boards of pharmacy [14]. Findings demonstrated that, in addition to the three states where technicians were already allowed to immunize, nine other states did not expressly prohibit this advanced technician role [14].

Doucette and Schommer recently assessed pharmacy technician willingness to undertake new advanced roles and identified variables that could improve this willingness [15]. Administering immunizations was found to be one of the tasks technicians were least willing take on, but education and support from the pharmacy team were found to be variables that were most associated with improving willingness to perform these tasks [15]. The American Pharmacist’s Association recently began offering an immunization training program specifically for pharmacy technicians [16]. McKeirnan and colleagues conducted a pilot study, training a small number of pharmacy technicians to administer immunizations, and found that immunizing technicians were competent, willing, and successful at this new role [17]. Time will tell how increased opportunity for technician education on this topic impacts technicians’ willingness to administer immunizations. 

One aspect of utilizing pharmacy technicians to administer immunizations that has not been explored is the impact on pharmacy workflow. Bertsch and colleagues showed that pharmacists who supervise immunizing technicians are supportive of this role, would encourage more technicians to become immunization-trained, and believe having immunizing technicians has increased the number of immunizations given at the pharmacy [18]. However, more information about how immunizing technicians are utilized in workflow may encourage the expansion of this new advanced role. This in mind, the objective of this work is to gather more information and provide additional insight on the topic of immunization and pharmacy workflow. Specifically, understanding how immunizing technicians are utilized in workflow, how often technicians are administering immunizations, and determining existing barriers to utilizing technicians in immunization workflow are the goals of this research. 

## 2. Materials and Methods 

This research was designed as a two-phase qualitative descriptive study utilizing key informant interviews. The first phase was conducted in 2017 [18]. Pharmacists within one pharmacy chain were contacted to participate in a key informant interview. These pharmacists supervised the first group of immunizing technicians in the United States trained during the pharmacy technician immunization training pilot project conducted by McKeirnan and colleagues in 2016. A description of the 2016 training pilot project can be found elsewhere [17]. The pharmacists from this chain were chosen as participants because they had more experience supervising immunizing technicians than any other pharmacists in the United States at the time of the first phase due to their technicians’ participation in this pilot project. 

### 2.1. Study Phase One

During phase one, pharmacists from the Albertsons corporation were contacted if at least one pharmacy technician employed at their pharmacy was included in the 2016 training pilot project. Pharmacy technicians were trained in December of 2016; key informant interviews were conducted six months later (May 2017). Researchers aimed to understand the perspective of these supervising pharmacists when incorporating trained immunizing technicians into their pharmacy. Two sets of interview questions were developed during this project. One was related to pharmacists’ perceptions of the implementation (“perceptions questions”), and the second was created to inquire further into the impact on pharmacy workflow (“workflow questions”). Rogers’ Diffusions of Innovations theoretical framework was utilized to help researchers disseminate this novel information to others wishing to adopt this practice. The perception questions were coded to the Five Stages of the Adoption Process and the workflow questions were coded to Rogers’ 5 Factors [19]. Rogers’ 5 Factors are intrinsic characteristics of innovation that influence the decision regarding whether to adopt a new idea or innovation [19].

The perception questions were created to specifically target supervising pharmacists’ initial trust and utilization of immunizing technicians, perceptions about the training program, and recommendations to other pharmacists who are considering having immunization-trained technicians [18]. The workflow questions focused on embedding immunizing technicians into pharmacy workflow. These workflow questions best paired with the theoretical framework of Rogers’ 5 Factors in the Diffusion of Innovation [19]. In this theoretical framework, Rogers defines five characteristics of innovation that influence the adoption or rejection of an innovation by an individual. Rogers describes these characteristics as interrelated but conceptually distinct. These five characteristics include:Relative Advantage: the degree to which the new option is improved (or not) over a previous version or standard.Compatibility: the degree to which the new option fits in with existing values and needs and can be assimilated into the potential adopter’s life.Complexity or Simplicity: the degree to which an innovation is perceived as difficult or simple to understand and implement.Trialability: the degree to which an innovation may be trialed and customized during the implementation process.Observability: the degree to which the results are visible to adopters or others.

These characteristics were utilized to develop five of the seven survey questions, as displayed in Table 1.

The key informant interview question script was developed by the primary investigator (TB), a licensed pharmacist who had experience providing the WSU Pharmacy Technician Immunization Training program but was not involved in training any of the technicians who were supervised by the participating pharmacists. The interview questions were peer-reviewed by colleagues. Key informant interviews were offered to supervising pharmacists at all 20 Albertsons pharmacies in Idaho State that had at least one pharmacy technician who attended the initial 2016 immunization training program. Initially, each pharmacy was emailed a copy of both sets of study interview questions by the district clinical coordinator. The intent of emailing the questions ahead of the phone interviews was to provide opportunity for the participants to give the questions thoughtful consideration and to minimize disruption of workflow. 

The primary investigator (TB) called each pharmacy during normal business hours and both sets of interview questions were asked. If the participant was not available or did not have time to answer all of the questions, the researcher offered to call back at a more convenient time. Pharmacists who were willing and available to participate were informed that participation was voluntary and that the decision whether or not to participate would not be shared with pharmacy management. Pharmacists were also told the conversation would be audio-recorded but individual participant names and locations would be removed prior to analysis and the dissemination of results. These study methods were found to be exempt from the need for review by the Washington State University Institutional Review Board (WSU IRB, #16030).

After all of the interviews were completed, the audio files were transcribed using an online transcription service (https://www.rev.com/) and redacted of information that could identify the participant or specific store. The transcriptions were reviewed, and each set of questions was coded separately using qualitative coding methods. Qualitative coding procedures, as described by Miles et al., were performed by two researchers [20]. First-level coding, the systematic labeling of items or concepts that appeared repeatedly in the text, was completed by hand independently. The researchers then met to discuss and cluster the codes into higher-level categories, performing second-level coding to create themes. Disagreements were resolved through discussion. Results from the perception questions demonstrated that saturation had been reached on this topic and a manuscript was published [18]. After reviewing the results from the workflow questions, researchers decided that saturation had not been reached on this topic. In order to achieve a more in depth understanding of the integration of immunizing pharmacy technicians into workflow further research would need to be conducted.

### 2.2. Study Phase Two

Phase two was conducted in January and February of 2020. Initially, the intent was to contact all 19 pharmacies that had participated in phase one of the study. However, permission was only given to contact five pharmacies located in northern Idaho. The same key informant workflow interview questions as shown in Table 1 were utilized again with the addition of one question previously included in the perceptions question list: “What percentage of the time does your technician(s) administer the immunizations?” The researchers believed comparing previous results of this question would help future adopters discern how immunizing technician utilization had varied over time in the workflow. 

Key informant interviews were held by the same researcher (TB) who conducted the initial interviews in 2017. Individual pharmacies were called during normal business hours with consideration given to which times of the day would likely be less busy. Willing participants were informed this was a follow-up study for pharmacists who supervised immunizing pharmacy technicians. Following the same methods as phase one, all interviews were audio-recorded and transcribed, and identifying information was redacted. First-level and second-level coding were performed using the same methods described during phase one. After coding was complete, 2017 interview findings were compared with 2020 findings and mapped to corresponding domains in Rogers’ 5 Factors of the Diffusion of Innovation [19]. An integral component associated with Rogers’ theory of innovation is time; innovations need to be tested over time in order to determine value [19]. The second set of interviews were conducted three years later with the same set of pharmacies, minus those who were unable to participate. Researchers determined that a saturation point in thematic findings had been met, as the outcomes were similar enough between the two sets of interviews. The 2020 responses repeated the majority of comments recorded previously in 2017. The research yielded no new data after this specific lapse in time.

## 3. Results 

During phase one in 2017, 19 pharmacists, each from a separate individual pharmacy within the same chain, agreed to participate in the key informant interviews. One pharmacist declined to participate during the interview. During phase two in 2020, five of the original pharmacies that were contacted had a pharmacist who was willing to participate, and all five pharmacies still employed pharmacy technicians who administered immunizations. Participant demographics from 2017 [18] and 2020 are included in Table 2 and Table 3. 

Qualitative analysis led to the following themes mapped back to each of Rogers’ 5 Factors, which are intrinsic components associated with the adoption of an innovation. Specifics on Rogers’ 5 Factors, relative advantage, compatibility, complexity or simplicity, trialability, and observability are described within the Methods section.

### 3.1. Factor 1: Relative Advantage

The relative advantage domain had one theme: improved flexibility towards creating a continuous workflow associated with immunizations administered within the pharmacy. This was reported by pharmacists in 2017 and confirmed by similar responses from 2020.

**Theme** **1.**
*Pharmacists believe having immunizing technicians improved the pharmacy workflow flexibility involved with immunizations.*

*“For us specifically, the way that our system’s designed it’s been great during busy times. We’ll send the technician in to do the immunizations so that we can continue to keep the workflow moving in the pharmacy. In that respect it’s been really positive, and just having the extra person in the store that can give [immunizations]. It’s going to be great during flu season.” (2017 Pharmacist 1)*

*“The pharmacist doesn’t feel too overburdened, especially during flu season and they feel like they can defer some of those responsibilities to somebody else.” (2017 Pharmacist 2)*

*“It’s kind of amazing because if you are stuck counseling or verifying something or on the phone due to an issue, [the immunizing technicians] are able to help with the immunization aspect of it.” (2020 Pharmacist 5)*

*“It helps free me up. As long as I’m trusting [my technician] to do the shot because I feel like I’m getting pulled in a million directions, it just helps and is paying off.” (2020 Pharmacist 1)*



### 3.2. Factor 2: Compatibility

Compatibility is described as the assimilation of an innovation into a particular model. In this model the pharmacy technicians’ compatibility with the newly provided service was highlighted by their supervising pharmacists. 

**Theme** **2.**
*Pharmacists believe in choosing a confident and friendly technician to provide immunizations.*

*“I think technicians who are very people-friendly will do better doing this, technicians who can go back and talk to the patient and put them at ease. It helps to have them be the person who starts them at the window and actually gives the shot, too. I think that having that person through the process helps.” (2017 Pharmacist 3)*

*“I have a technician who is confident in herself…you’ve got different personalities, and she’s definitely one of the appropriate personality types for that.” (2017 Pharmacist 4)*

*“I would only do it if a technician is a go-getter and wants to do it. I would never put a technician on the spot if they weren’t comfortable with it. I would never want them to have to feel that they were being pushed into doing it because they were a technician.” (2020 Pharmacist 4)*

*“I think a lot of it’s the technician’s personality. I try and pick technicians who are really comfortable with it, and if they’ll own it, those are the ones who are going to be most successful at it. That’s what I would say to look for, when you choose technicians to do it look for ones that that’s going to fit their personality type.” (2020 Pharmacist 3)*



### 3.3. Factor 3: Complexity or Simplicity

The complexity or simplicity of introducing a newly immunization-trained pharmacy technician into the workflow should be considered by stakeholders. The level of effort required to train and observe does not immediately improve workflow. However, after comfort is established by the supervising pharmacist the level of complexity for each immunization is reduced.

**Theme** **3.**
*Supervising pharmacists believe the innovation of having a technician capable of immunizing within the workflow helps to better prioritize their time.*

*“Having to stop workflow to go and give a whole family of five people flu shots tends to be difficult. Once [my technicians] were able to [immunize], it saves a lot of time. It makes it so that workflow doesn’t have to stop if I’m the only pharmacist here. I can say we need an injection and we keep on rolling. (2017 Pharmacist 5)*

*“Having that extra hand if we need it is very helpful, so we don’t get overrun. Because I don’t know about you, but in my flu shot season, we’re doing upwards of 40 a day.” (2020 Pharmacist 5)*

*“Especially during flu season, there are times where we have multiple people getting immunizations. And so during those times, both myself and the technician will be giving shots, at the same time.” (2020 Pharmacist 2)*

*“I’m the pinch point, and again, I mean it just depends what’s going on, and a lot of times, if I’m busy verifying or doing something, I’ll just ask the technician, ‘would you please give this person their immunization?’ ” (2020 Pharmacist 1)*



### 3.4. Factor 4: Trialability

Trialability helps stakeholders to determine how easily an innovation can be adopted. By testing for this factor, adopters can anticipate certain pitfalls to avoid. After implementation and trial, the barriers of the new innovation should be considered to determine whether stakeholders should adopt the new practice. Ultimately, one recurrent theme was highlighted by the key informant interviews.

**Theme** **4.**
*Pharmacists wanted more immunization-trained pharmacy technicians in their pharmacies.*

*“I mean that was the biggest roadblock is that we had individuals that we wanted to get certified, but the program just wasn’t available to be able to get them to do that.” (2020 Pharmacist 2)*

*“Yeah. I think everybody’s going to eventually see the benefit in it. We’ve got a second technician going through the training course next week. We’ll have two technician immunizers in our pharmacy here soon. It’s pretty easy to see the benefits of it though when you look at the workflow.” (2017 Pharmacist 2)*

*“[The biggest challenge is] the ease of getting the training.” (2020 Pharmacist 5)*



### 3.5. Factor 5: Observability

Transparency and the observability of the opinions of supervising pharmacists or early adopters provide an effective way to create either positive or negative communication channels to drive decisions. Pharmacists communicated that having an immunization-trained pharmacy technician as part of the workflow was positive. 

**Theme** **5.**
*Pharmacists as observers were initially hesitant, then accepting of the added member to the immunization team.*

*“So I only have one full time that is eligible that went through the training so pretty much any vaccination that came up, I pretty much put a tech to go vaccinate. So I would still have to check the prescription and go over the paperwork first but…We would go, prepare all the gloves and everything and get all the side work done, and then go administer the vaccination.” (2017 Pharmacist 7)*

*“Take advantage of it. I mean, I’m sure that some of them might feel hesitant allowing the tech to be able to do that, because we’ve all [thought], ‘Oh, no, it’s the pharmacist’s job’. But you have to jump onboard and trust your teammates.” (2020 Pharmacist 5)*



## 4. Discussion

Rogers’ 5 Factors should help community pharmacy stakeholders determine the rate at which this innovation should be adopted [19]. Reflecting on the results and themes produced in terms of relative advantage, compatibility, complexity, or simplicity, trialability, and observability is necessary to make an informed decision. In summation, the perception of the supervising pharmacists was that having an immunizing technician improved workflow and allowed for improved time prioritization. However, some concerns were highlighted: The choice of the appropriate technician to receive the training was considered important, and the current low offer of immunization training was perceived to be a barrier towards implementation. In addition, initial hesitation and temporarily increased workload were expressed as challenges by supervising pharmacists who introduced an immunizing technician into the workflow. 

Pharmacy technicians have a palpable impact on community pharmacy workflow, and advanced technician roles have been shown to positively affect technician job satisfaction [8]. In addition, patient care aspects in pharmacy technician roles contribute to increased self-actualization [21]. Providing pharmacy technicians with perceivably meaningful activities, such as involving them in patient care, can benefit their work performance [22]. Fostering innovations that can produce new workplace environments such as this can improve the traditional community pharmacy paradigm. 

There were limitations to this research. During phase 1, interview questions were emailed to staff pharmacists prior to the interviews with the goal of providing opportunity for thoughtful consideration of the questions and minimizing disruption to workflow. While these goals may have been achieved, providing the questions ahead of time may have created bias since the participants had the opportunity to provide responses that were formulated rather than giving the reactionary responses expected during interviews where the participant cannot prepare ahead of time. 

Although researchers were not able to contact and interview all 20 pharmacies in phase 2, researchers believed saturation was reached after conducting the 2020 interviews because the 2020 responses repeated the majority of the comments recorded in 2017, and because all five of the 2020 interviews yielded similar results. One pharmacy in 2020 had no current immunizing pharmacy technician, which was a recent change that had occurred less than one month prior to the interview (Table 3). As pharmacy technician immunization administration becomes more widespread, conducting similar research on pharmacy workflow with a larger and more diverse key informant group could lead to different results. 

This project was conducted in one state and within one pharmacy chain. Information about individual pharmacy prescription volumes, number of patients, and number of employees was not available to the research team, but could provide valuable insight into how immunizing technicians are utilized in workflow in stores with varying degrees of staff support and time available to engage with each patient. This pharmacy chain was a very early adopter of this new advanced technician role and chain leadership was very supportive. Results of similar work in a chain where pharmacists or pharmacy leadership are less supportive of immunizing technicians would likely lead to different results. Results may also differ with pharmacists that are not comfortable with immunizing patients. All the pharmacists interviewed in this study already administered immunizations before immunizing technicians were added into the workflow. Additionally, because of the way the research was conducted, the pharmacists interviewed in 2017 were not necessarily the same pharmacists who were interviewed in 2020. Conducting a similar project longitudinally with the same subset of pharmacists may provide a more detailed picture of the impact on workflow. 

There is still much research to be done on this topic. Although pharmacists’ perceive an increase in the number of immunizations administered [18] when immunizing technicians are integrated into pharmacy workflow, actual immunization data comparing stores with immunizing technicians to similar stores without immunizing technicians would strengthen these results. Similarly, pharmacists perceive that workflow is improved and pharmacist time is saved by utilizing technicians to administer immunizations, but conducting a study similar to that of Fleagle and colleagues in 2019 where workflow hours were analyzed would lead to definitive results about the amount of time saved for pharmacists. Additionally, the training available for pharmacy technicians to learn to administer immunizations is being expanded from a small program at Washington State University [23] to a program offered on a national level through the American Pharmacists Association [16]. Since one of the challenges identified by the study pharmacists was lack of availability of the training program because they had willing technicians who were not able to attend to date, the implications of broader access to the program have not yet been realized. This training expansion will lead to a multitude of additional research opportunities as technicians in more geographic regions of the country with varying needs for additional immunizers begin undertaking this role.

## 5. Conclusions

Rogers’ 5 Factors from the Diffusion of Innovation provide insight into ideal characteristics for encouraging adoption of an innovation. Immunizing is a relatively new role for pharmacy technicians, and consideration of factors that can encourage and ease implementation into pharmacy workflow can aid in future application. The findings of this study included themes such as: (1) Pharmacists’ perceived improvement in workflow flexibility; (2) The choice of the correct technician to immunize within the pharmacy; (3) Pharmacists’ perceived improved workflow time prioritization; (4) Limited available training as a barrier to implementation; and (5) The initial apprehension and later acceptance of pharmacists with respect to the innovation. Pharmacists are able to focus on the task at hand rather than facing interruptions and delays in checking prescriptions caused by providing walk-in immunizations. The biggest barrier identified by the participant pharmacists was the challenge of getting more technicians trained. Technicians were interested in immunizing and pharmacists supported them in becoming immunizers, but the training was not being offered in their area as frequently as they would prefer. As technician immunization administration spreads beyond early adopter states, further research into the impact on pharmacy workflow is needed.

## Figures and Tables

**Table 1 pharmacy-08-00071-t001:** Key informant interview questions.

Key Informant Interview Questions	Rogers’ 5 Factors Characteristic
(1) How long have you been working as a pharmacist?	Demographics
(2) How many immunizing technicians do you currently have at your pharmacy?	Complexity or Simplicity
(3) How has the addition of immunizing technicians impacted pharmacy workflow?	Trialability
(4) What did you find was the best way to utilize your pharmacy technician(s) when administering immunizations?	Compatibility
(5) Describe all barriers or challenges that you felt you had to overcome in introducing technicians in the immunization workflow.	Complexity or Simplicity
(6) Would you recommend having immunizing technicians to other pharmacists? Why or why not?	Relative Advantage (or lack thereof)
(7) What percentage of the time does your technician(s) administer the immunizations? *	Observability
(8) Do you have any additional feedback that was not addressed by this questionnaire?	General

* Question used from 2017 [18] and 2020 only.

**Table 2 pharmacy-08-00071-t002:** Demographics for 2017 [18] and 2020.

**2017 Demographics [18]**			
**Pharmacist ID and Store Number**	**Gender**	**Would Recommend Technician Immunization Training**	**Percentage of Time the Technician Administers Immunizations vs. the Pharmacist**
1	M	Y	70%
2	F	Y	50%
3	F	Y	85%
4	F	Y	75%
5	F	Y	80%
6	M	N	100%
7	M	Y	75%
8	F	Y	60%
9	M	Y	100%
10	F	Y	80%
11	M	Y	100%
12	M	Y	10%
13	F	Y	80%
14	F	Y	50%
15	M	Y	95%
16	M	Y	70%
17	F	Y	50%
18	F	Y	100%
19	M	Y	100%
			Est Average: 74%
**2020 Demographics**			
**Pharmacist ID and Store #**	**Gender**	**Would Recommend Technician Immunization Training**	**Percentage of Time the Technician Administers Immunizations vs. the Pharmacist**
1	F	Y	50%
2	M	Y	80%
3	M	Y	100%
4	F	Y	70%
5	F	Y	35%
			Est. Average: 67%

**Table 3 pharmacy-08-00071-t003:** Key informant interview demographics continued.

2020 Demographics		
Pharmacist ID and Store Number	How long have you been working as a pharmacist?	Current Immunizing Technicians
1	16 years	0 *
2	N/A	2
3	16 years	1
4	N/A	1
5	2 years	1

* No active immunizing technician for one month.

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
