# Peer review of "Perceived Benefit of Immunization-Trained Technicians in the Pharmacy Workflow"

_pharmacy, 2020, doi:10.3390/pharmacy8020071_

Round 1

Reviewer 1 Report

This paper reports on a qualitative study performed on a small sample of pharmacists about the perceived benefit of immunization trained technicians in the pharmacy workflow.

This study has some flaws that undermine the results presented.

Some comments follow:

Check for spelling:

Line 14 – spelling “Findings”

Line 111 – “difficult”

Line 142 – “as described”

Line 277 – “felt that having”

Methods:

Line 127 – sending the questions beforehand could bias the answers (i.e. giving more time to think on the “appropriate” answer). Did the researchers account for that bias? More information on this would be nice.

Why the choice of pharmacists from a specific corporation (i.e. Albertsons)?

Results:

Phase two was conducted with a very small number of participants, thus hindering the results. It seems that there was a massive loss of interest by the corporation. They were not even the same persons as the authors recognize in the limitations sections. Researchers should have planed for that initially.

Regarding the time technicians were providing immunizations, did you find any increase? Were the pharmacies in 2020 the same as in 2017? It is not clear.

Line 294 -295 – I disagree with the finding that “saturation was reached after 2020 interviews”. Nothing in your data points to that saturation.

All the limitations the authors state point to the low interest of this research.

Author Response

Reviewer 1

This paper reports on a qualitative study performed on a small sample of pharmacists about the perceived benefit of immunization trained technicians in the pharmacy workflow.

Thank you for your feedback, we appreciate your input and do acknowledge that this sample size is perceivable as small.  

To reflect on this, we hope the readers can visualize that this a longitudinal qualitative study, conducted with the only pharmacists in the United States that had supervised immunization trained technicians.

Check for spelling:

Line 14 – spelling “Findings”

Line 111 – “difficult”

Line 142 – “as described”

Line 277 – “felt that having”

Changes made, thank you.

Methods

Line 127 – sending the questions beforehand could bias the answers (i.e. giving more time to think on the “appropriate” answer). Did the researchers account for that bias? More information on this would be nice.

We agree. Here is the addition made about this to the limitations section: “There were limitations to this research. During phase 1, interview questions were emailed to staff pharmacists prior to the interviews with the goal of providing opportunity for thoughtful consideration of the questions and minimizing disruption to workflow. While these goals may have been achieved, providing the questions ahead of time may have created bias since the participants had the opportunity to provide responses that were formulated rather than the reactionary responses expected during interviews where the participant cannot prepare ahead of time.”

Why the choice of pharmacists from a specific corporation (i.e. Albertsons)?

Thanks for the question.  We added the following sentence to the first paragraph of the Materials and Methods section to clarify this point: ” The pharmacists from this chain were chosen as participants because they had more experience supervising immunizing technicians than any other pharmacists in the U.S. at of the time of the first phase due to their technicians’ participation in this pilot project.”

Results:

Phase two was conducted with a very small number of participants, thus hindering the results. It seems that there was a massive loss of interest by the corporation. They were not even the same persons as the authors recognize in the limitations sections. Researchers should have planed for that initially.

Thank you for the feedback.  When we initially designed the study we believed it would involve interviews during one time period.  However, we decided a second phase was needed for this topic.  At that point we regretted our decision not to identify the pharmacists which would have allowed us to determine how many of the pharmacists from the phase one were also involved in phase two and how those comments compared to the comments of pharmacist who were not involved in phase one.  If we conduct a similar study in the future, we will keep this in mind.

Regarding the time technicians were providing immunizations, did you find any increase? Were the pharmacies in 2020 the same as in 2017? It is not clear.

Objective findings involving the specific number of immunizations provided by the pharmacy technicians were not tracked. Anecdotal use of technician optimization within the immunization workflow was included. The pharmacies interviewed were from the same pool. In the materials and methods line 156- 158 states, “Initially, the intent was to contact all 19 pharmacies that had participated in phase one of the study. However, permission was only given to contact five pharmacies located in northern Idaho.”

To be helpful to the reader it was added in the results (Line 178-179), “During phase two in 2020, five of the original pharmacies that were contacted had a pharmacist who was willing to participate, and all five pharmacies still employed pharmacy technicians who administered immunizations.”

Line 294 -295 – I disagree with the finding that “saturation was reached after 2020 interviews”. Nothing in your data points to that saturation.

Thank you for pointing this out. 

We added the following to the methods section to clarify our rationale: “After reviewing the transcripts from the 2020 interviews and comparing them with the 2017 interviews, the researchers believed saturation had been reached.  This conclusion was determined since the 2020 responses repeated the majority of the comments recorded in 2017 and because all five of the 2020 interviews yielded similar results.”

Additionally, in the discussion, we added the following to clarify this point: “Although researchers were not able to contact and interview all 20 pharmacies in phase 2, researchers believed saturation was reached after conducting the 2020 interviews because the 2020 responses repeated the majority of the comments recorded in 2017 and because all five of the 2020 interviews yielded similar results.”

Reviewer 2 Report

The paper is Clear, the Method well described and the results well presented. I have no major suggestions for improvement. The only thing is that, either in Methods or in limitations, there should be some mention of which type of stores where involved in the Research. The Authors say that the stores were in the same state and the same Chains, but was the type of store (size, number of employees, location - city centre? country side? in a street or in a shopping mall?) different or the same? This can be relevant because the type of customers, relation pharmacist-customer, and time dedicated to each customer is usually different in different type of pharmacies.

There is a typo in line 188 last Word, and at page 277 a 'having' should be deleted (second last Word).

Author Response

Reviewer 2:

The paper is Clear, the Method well described and the results well presented. I have no major suggestions for improvement. The only thing is that, either in Methods or in limitations, there should be some mention of which type of stores where involved in the Research. The Authors say that the stores were in the same state and the same Chains, but was the type of store (size, number of employees, location - city centre? country side? in a street or in a shopping mall?) different or the same? This can be relevant because the type of customers, relation pharmacist-customer, and time dedicated to each customer is usually different in different type of pharmacies.

This is an important point and I wish we had more information on this topic.  I hope in the future we will have the opportunity to conduct a study looking at the differences in immunizing technician utilization in workflow at stores where these variables could be compared.  It would also be beneficial when we have actual immunization data rather than perceptions to make the comparison more robust.  An addition was made to our limitations section to address this: “Information about the individual pharmacy prescription volumes, number of patients, and number of employees were not available to the research team, but could provide valuable insight into how immunizing technicians are utilized in workflow in stores with varying degrees of staff support and time available to engage with each patient.”

There is a typo in line 188 last Word, and at page 277 a 'having' should be deleted (second last Word).

Thanks, changes made as suggested.

Round 2

Reviewer 1 Report

Thanks for the great job in improving the paper. 

Some minor spell checking in:

Line 85 - "at of the time"

Line 178 - "fiveof the original"

Beyond this, there still remains the question about saturation of results. I feel that 5 interviews are not enough to determine that a saturation has been reached. However, I accept your point of view, stated in Lines 168 - 171. I feel that this paragraph should be rephrased. In the methods you should just define how and when saturation was reached.

Author Response

Thanks for the great job in improving the paper.

#1. Some minor spell checking in:

Line 85 - "at of the time"

Line 178 - "fiveof the original"

-Made sure to fix these, thank you very much!

Beyond this, there still remains the question about saturation of results. I feel that 5 interviews are not enough to determine that a saturation has been reached. However, I accept your point of view, stated in Lines 168 - 171. I feel that this paragraph should be rephrased. In the methods you should just define how and when saturation was reached.

-We pondered this question and thought of how to better represent or methods. Changed around the paragraph structure so that was not quite as clunky, and added (lines 171-177), "An integral component associated with Rogers’ theory of innovation is time, innovations need to be tested over time in order to determine value [19]. The second set of interviews were conducted three years later with the same set of pharmacies, minus those who were unable to participate. Researches determined a saturation point in thematic findings had been met as the outcomes were similar enough between the two sets of interviews. The 2020 responses repeated the majority of comments recorded previously in 2017. The research yielded no new data after this specific lapse in time."

Our thoughts were that this is a very valid point - and the nature of meeting data saturation can be somewhat obscure. We felt the best we could do as a team is be as transparent as possible here, by telling the audience that saturation was met at the discretion of the researchers. Hopefully I was able to define the 'how' and 'why' of your question with this. Thank you for your time.